# Transcriptome Profile Analysis of Intestinal Upper Villus Epithelial Cells and Crypt Epithelial Cells of Suckling Piglets

**DOI:** 10.3390/ani12182324

**Published:** 2022-09-07

**Authors:** Lijun Zou, Yirui Shao, Yinfeng Xu, Yuliang Wu, Jian Zhou, Xia Xiong, Yulong Yin

**Affiliations:** 1Laboratory of Basic Biology, Hunan First Normal University, Changsha 410205, China; 2Key Laboratory for Agro-Ecological Processes in Subtropical Regions, Institute of Subtropical Agriculture, The Chinese Academy of Sciences, Changsha 410125, China

**Keywords:** suckling piglet, intestinal epithelial cells, crypt-villus axis, cell renewal, transcriptome

## Abstract

**Simple Summary:**

To investigate the genetic reprogramming that drives intestinal epithelial cells (IECs) maturation along the crypt-villus axis, entCerocytes were sequentially isolated from the villus tip to the crypts of porcine small intestine. The present study obtained the intestinal upper villus epithelial cells (F1) and crypt epithelial cells (F3) of 21-day suckling piglets using the divalent chelation and precipitation technique. By mRNA high-throughput sequencing analysis of F1 and F3, it was shown that a total of 672 unigenes were differentially expressed between F1 and F3, including 224 highly expressed and 448 minimally expressed unigenes. The greatest number of differentially expressed genes enriched in signal transduction, e.g., Wnt, Hippo, TGF-beta, mTOR, PI3K-Akt, and MAPK signaling pathways, were closely related to the differentiation, proliferation, maturation and apoptosis of IECs. Our results provide important information for identifying new regulators of IECs differentiation and maturation. Moreover, they can also provide insights into the regulatory mechanisms underlying intestinal epithelial cell renewal and the rapid repair of intestinal mucosal oxidative damage.

**Abstract:**

It is well known that the small intestinal epithelial cells of mammals rapidly undergo differentiation, maturation, and apoptosis. However, few studies have defined the physiological state and gene expression changes of enterocytes along the crypt-villus axis in suckling piglets. In the present study, we obtained the intestinal upper villus epithelial cells (F1) and crypt epithelial cells (F3) of 21-day suckling piglets using the divalent chelation and precipitation technique. The activities of alkaline phosphatase, sucrase, and lactase of F1 were significantly higher (*p* < 0.05) than those of F3. To explore the differences at the gene transcription level, we compared the global transcriptional profiles of F1 and F3 using RNA-seq analysis technology. A total of 672 differentially expressed genes (DEGs) were identified between F1 and F3, including 224 highly expressed and 448 minimally expressed unigenes. Functional analyses indicated that some DEGs were involved in the transcriptional regulation of nutrient transportation (*SLC15A1*, *SLC5A1*, and *SLC3A1*), cell differentiation (*LGR5*, *HOXA5* and *KLF4*), cell proliferation (*PLK2* and *TGFB3*), transcriptional regulation (*JUN*, *FOS* and *ATF3*), and signaling transduction (*WNT10B* and *BMP1*), suggesting that these genes were related to intestinal epithelial cell maturation and cell renewal. Gene Ontology (GO) enrichment analysis showed that the DEGs were mainly associated with binding, catalytic activity, enzyme regulator activity, and molecular transducer activity. Furthermore, KEGG pathway analysis revealed that the DGEs were categorized into 284 significantly enriched pathways. The greatest number of DEGs enriched in signal transduction, some of which (Wnt, Hippo, TGF-beta, mTOR, PI3K-Akt, and MAPK signaling pathways) were closely related to the differentiation, proliferation, maturation and apoptosis of intestinal epithelial cells. We validated the expression levels of eight DEGs in F1 and F3 using qRT-PCR. The present study revealed temporal and regional changes in mRNA expression between F1 and F3 of suckling piglets, which provides insights into the regulatory mechanisms underlying intestinal epithelial cell renewal and the rapid repair of intestinal mucosal damage.

## 1. Introduction

The mammalian small intestine is not only an organ for food digestion and nutrient absorption, but also a critical component of the endocrine and immune systems [1,2]. Furthermore, it has to withstand intestinal microbial colonization, extreme pH variations and mechanical abrasion [3]. To fulfill these functions, the intestinal epithelium has become one of the most rapidly self-renewing tissues (approximately 5~7 days) in mammals [4]. The mammalian intestinal epithelium is composed of a single layer of columnar epithelium, which is mainly divided into two different parts: the differentiation area and the proliferation area. The differentiated epithelial cells are mainly found on the villus, and the proliferative epithelial cells are mainly located in the crypt. The crypt mainly consists of intestinal stem cells (ISCs) and transit amplifying (TA) cells, in which TA cells can rapidly proliferate and migrate to the differentiation area along the crypt-villus axis (CVA). Mature IECs mainly contain four types of cells: absorptive epithelial cells (enterocytes) and three secretory cell lineages (enteroendocrine cells, goblet cells and Paneth cells) [3]. Since absorptive epithelial cells are rich in digestive enzymes and nutrient transporters, they are mainly responsible for the absorption of nutrients, which account for more than 95% of villous cells [5]. In addition, secretory IECs are closely related to the intestinal immune barrier. ISCs undergo proliferation, which produces a population of differentiation lineage-specific cells that are rapidly renewed. Enterocytes, enteroendocrine cells and goblet cells migrate along the CVA from the crypt to the tip of the villus, where they are exfoliated into the gut lumen via apoptosis, whereas Paneth cells migrate towards the crypt and reside at the bottom of the crypt, which are tightly regulated processes [5].

The proliferation, differentiation, and apoptosis of IECs play a key role in the damage repair mechanism after immune response elicitation in the intestine, for restoring the intestinal barrier function and homeostasis. It is well known that the differentiation of ISCs depends on external signals from epithelial and non-epithelial cells (T helper cells and stromal cells). A number of signal pathways and factors that regulate the renewal of IECs in the intestine have been identified, e.g., Wnt, Notch, EGF, and CDX2. However, the molecular mechanisms involved in regulating these interactions remain largely undefined.

To improve the production efficiency of modern pig breeding, early weaning of piglets has become a common production method for large-scale breeding, and generally takes place at the age of 21 days. A series of studies has shown that early weaning can cause villus height shortening and shedding, crypt hyperplasia, and increase the expression of apoptosis-related genes in the small intestinal epithelium of piglets [6,7]. The small intestine is the most crucial organ associated with the growth performance of piglets. Numerous studies have focused on the nutritional regulation and molecular mechanisms of IEC injury repair after weaning in piglets. However, little is known about the renewal of IECs of suckling piglets at the genomic level. Therefore, the objective of the present study was to investigate the hierarchical profiles of mRNA expression of enterocytes along the CVA in the jejunum of suckling piglets by RNA-seq. Overall, our results were valuable for understanding the genomic information and the molecular mechanism of the renewal of IECs in suckling piglets. In addition, the present study provides important information for identifying new regulators of IEC differentiation and maturation.

## 2. Materials and Methods

All procedures using piglets were approved by the Animal Care and Use Committee of the Institute of Subtropical Agriculture at the Chinese Academy of Sciences.

### 2.1. Animal and CVA Cell Isolation

A total of five 21-day sucking piglets (Duroc × [Landrace × Yorkshire]) were used in the present experiment. Piglets were given a general anesthetic with an intravenous injection of sodium pentobarbital solution (40 mg/kg body weight) in the ear and then euthanized. The CVA epithelial cells of jejunum were isolated using the intestinal chelating precipitation method, as we previously described [8]. Briefly, the divided mid-jejunum segments (approximately 20 cm) were washed thoroughly with 4 °C normal saline solution, and then incubated with an oxygenated PBS buffer for 30 min at 37 °C (step 1). The PBS was poured out, and the jejunum segments were subsequently incubated with 200 mL of a oxygenated chelating buffer (0.5 mmol/L dl-β-hydroxybutyrate sodium salt, 0.5 mmol/L DTT, 5 mmol/L Na_2_EDTA, 2.5 mmol/L l-glutamine, 10 mmol/L HEPES pH 7.4, 0.25% BSA, 2.5 mmol/L d-glucose, oxygenated with an O_2_/CO_2_ mixture (19:1, *v*/*v*), pH 7.4) for 40 min at 37 °C, and the chelating buffers were collected in 50 mL centrifuge tubes and centrifuged (400× *g*) for 10 min at 4 °C. This procedure was repeated three times to yield three “cell fractions” (F1 to F3) (step 2). The cells collected in step 2 were washed twice with an oxygenated cell suspension buffer (2.0 mmol/L MgCl_2_, 1.5 mmol/L CaCl_2_, 10 mmol/L HEPES, pH 7.4) and then centrifuged (400× *g*) for 10 min at 4 °C (step 3). The cells collected first (F1) and the cells collected last (F3) were the intestinal upper villus cells and crypt epithelial cells, respectively. The cells were immediately frozen in liquid nitrogen and then stored at −80 °C until further processing.

### 2.2. Measurement of Alkaline Phosphatase, Lactase, and Sucrase Activities

The alkaline phosphatase (ALP), lactase and sucrase activities of the cells were measured using their corresponding enzyme assay kit (Jiancheng Biotech, Nanjing, China) according to the manufacturer’s protocol. Briefly, cells were washed with PBS (three times) and incubated with 100 μL lysis buffer for 10 min at 4 °C, and then centrifuged at 12,000× *g* for 10 min at 4 °C. The supernatants were collected for measuring enzyme activity using specific substrate binding method. Absorbance (OD value) was read at 505 nm using a microplate reader (BioTek Instruments Inc., Winooski, VT, USA). The protein concentration of cells was measured using a BCA protein assay kit (Biyotime Biotech, Shanghai, China). ALP, sucrase, and lactase activities were normalized to protein concentration in parallel experimental plates (enzyme activity $=\frac{\mathrm{OD}\left(\mathrm{sample}\right)-\mathrm{OD}\left(\mathrm{blank}\right)}{\mathrm{OD}\left(\mathrm{standard}\;\mathrm{solution}\right)-\mathrm{OD}\left(\mathrm{blank}\right)}\times \frac{\mathrm{Concentration}\left(\mathrm{standard}\;\mathrm{solution}\right)}{\mathrm{Concentration}\left(\mathrm{sample}\;\mathrm{protein}\right)}$). The unit of enzyme activity is given as μmol product/min/mg protein.

### 2.3. Total RNA Isolation

The total RNA for the mRNA sequencing of F1 and F3 was extracted with the TRIZOL reagent (TaKaRa, Osaka, Japan), according to the corresponding manufacturer’s protocol. The integrity of the total RNA was assessed using the Agilent BioAnalyzer 2100 (Agilent Technologies, Santa Clara, CA, USA) and the purity (OD 260: OD 280 ≈ 2.0) and concentration of RNA were determined using a NanoDrop ND-2000 spectrophotometer (Thermo Fisher, Waltham, MA, USA).

### 2.4. Processing of High-Throughput Sequencing Data

The ten complementary DNA (cDNA) libraries were constructed using a TruSeq^®^RNA sample prep kit (Illumina, San Diego, CA, USA) according to its manufacturer’s protocol. In brief, the 2 µg of total RNA was prepared by diluting to 50 µL of nuclease free ddH_2_O. First, the total RNA was purified by mRNA enrichment or rRNA removal. Second, fragmentation of the RNA with an interrupt buffer, reverse transcription for cDNA generation, and then synthesis of cDNA two strands to form double-stranded DNA were carried out. Third, the double-stranded DNA was flattened and phosphorylated at the 5′ end, and the 3′ end formed a sticky end protruding from an “A”, which was connected to a bubbling linker with a protruding “T” at the 3′ end. Fourth, the ligation product was amplified by PCR with specific primers. Finally, the PCR product was thermally denatured into single-stranded DNA, and the single-stranded DNA was cyclized with a bridge primer to obtain a single-stranded circular DNA library.

After the quality control of the cDNA, pair-end sequencing of two libraries (F1 and F3) was performed via a BGISEQ-500 platform at the Beijing Genomics Institution (Shenzhen, China). To ensure the accuracy of de novo assembly, the clean reads were obtained by filtering reads with 5′ adaptor contaminants, reads without 3′ adaptor and the insert tag, poly-N sequences, and low-quality reads from raw data. After quality filtering, all clean reads were mapped onto the Sus scrofa genome (https://www.ncbi.nlm.nih.gov/genome/?term=pig (accessed on 15 October 2021)) using TopHat software Version 2.1.1 (University of Maryland, City of College Park, MD, USA). At the same time, the Q20 and mapped reads contents of the clean data were calculated. The high-quality clean reads were used for downstream analyses.

### 2.5. Differential Expression Analysis

The value of log2 Ratio (F3/F1) was normalized by using the local standard deviation of the mRNA expression level (count number) (according to the linear regression of all gene expression levels, taking a certain small area of the corresponding gene, and then calculating the corresponding standard difference), and then comparing and calculating the calculation. When the expression level of each mRNA is corrected, it is related to the expression distribution of all mRNA in each sample. The relative abundance of transcripts was calculated and normalized according to FPKM (fragments per kilobase per transcript per million mapped reads) for each unigene between F1 and F3. The *p*-values were adjusted using the Benjamini & Hochberg and Storey & Tibshirani methods, which are used in false discovery rate (FDR) for multiple hypothesis testing. The unigenes were defined as DEGs between two groups based on |log2 (fold change)| > 2 and FDR ≤ 0.001.

### 2.6. Cluster Analysis and Functional Annotation of DEGs

Gene Ontology (GO) enrichment analysis of the DEGs was implemented using the P heatmap software version 1.0.12 (AT&T Bell Laboratories, New York, NY, USA) in the R package version 1.10.0 (2014). Pathway analysis was employed to identify the critical signal pathways of the DEGs based on the resources from Kyoto Encyclopedia of Genes and Genomes (KEGG) Orthology Based Annotation System (KOBAS 2.0: http://kobas.cbi.pku.edu.cn/home.do (accessed on 18 October 2021)) software.

### 2.7. qRT-PCR Validation of the Transcriptome Results

The primers were designed with the use of Primer-Blast online software (https://www.ncbi.nlm.nih.gov/tools/primer-blast/index.cgi?LINK_LOC=BlastHome (accessed on 8 December 2021)) (National Institutes of Health, Bethesda, MD, USA) according to the porcine genome sequence, with *GAPDH* as the endogenous control. The primers were synthesized from TsingKe Biological Technology (Changsha, China) and are listed in Appendix A.

To detect the mRNA expression level, first-strand cDNA was synthesized using PrimeScript™ RT reagent Kit with genomic DNA Eraser (Takara, Osaka, Japan). qRT-PCR amplified was performed using the SYBR Green method and the model QuantStudio 5 Real-Time PCR System analyzer (Applied Biosystems, Foster City, CA, USA), as we described [8]. Briefly, the 20 µL assay solution reaction mixture contained 2 µL cDNA template, 10 µL SYBR Green mix (2×), 6.8 µL ddH_2_O, and 0.6 µmol/L each of forward and reverse primers (10 nmol/mL), and cycling conditions were 50 °C for 2 min, 95 °C for 10 min, followed by 40 cycles of 95 °C for 15 s, and 60 °C for 60 s. All qRT-PCR reactions yielded a single peak on the dissociation curve, indicating the specific amplification of the primers. Relative expression levels of target gene were calculated using the 2^−ΔΔCt^ method [9].

### 2.8. Statistical Analyses

All data are presented as the mean ± standard error of the mean (means ± SEM). All statistical analyses were performed using the SPSS 22.0 statistical package (SPSS Inc., Chicago, IL, USA). Data for enzyme activity and gene relative expression in F1 and F3 were analyzed by Student’s *t*-test. A value of *p* < 0.05 was taken to indicate statistical significance.

## 3. Results

### 3.1. Enzyme Activities of the Fractionation Procedure

We have isolated two cell fractions from jejunum CVA and validated the fractionation efficiency by ALP and disaccharidase activity (Figure 1). The specific activity of ALP increased 29.52-fold from 1.07 (F3) to 31.59 (F1) U/g protein, which is consistent with a previous study [7]. The specific activity of lactase and sucrase increased 16.72-fold from 0.53 (F3) to 8.86 (F1) U/mg protein and 21-fold from 0.17 (F3) to 3.57 (F1) U/mg protein, respectively. Taken together, these data indicate that differentiated villus upper cells and proliferating crypt cells of piglets can be efficiently isolated using the divalent chelation and precipitation technique.

### 3.2. Mapping of RNA-Seq Reads to Pig Genome

To identify critical genes that regulate IECs renewal, we carried out high-throughput sequencing of villus upper epithelial cells (F1) and crypt epithelial cells (F3). Approximately 23.96 million 150 bp paired-end reads were generated for each sample. After removing low-quality reads and short read sequences, a total of 23.88 million clean reads (89.28%) were obtained in each sample with Q20 (the percentage of bases whose mass value is greater than or equal to 20) bases more than 93.9% and 96.0% for F1 and F3, respectively. The ratio of clean reads for F1 and F3 that could be aligned to the genome was 88.35% and 90.21%, respectively, in which 57.16% and 63.18% of the clean reads were unique matches and 31.19% and 27.03% were multi-position matches (Table 1).

### 3.3. Differential Gene Expression Profiles of F1 and F3

To elucidate the molecular mechanism underlying the renewal of IECs, we first conducted the RNA-seq technology to compare the mRNA expression profile of F1 and F3 in suckling piglets. The analysis of clean data showed that there were 672 mRNAs that had different expression levels between F1 and F3 (Figure 2A). Of these, 224 were highly expressed (Appendix A) and 448 were minimally expressed (Appendix A) DEGs in F3 compared with the control group (F1) (Figure 2B). Some highly expressed genes, were Leucine Rich Repeat Containing G Protein-Coupled Receptor 5 (*LGR5*), Homeobox A5 (*HOXA5*), Jun Proto-Oncogene, AP-1 Transcription Factor Subunit (*JUN*), Fos Proto-Oncogene, AP-1 Transcription Factor Subunit (*FOS*), Early Growth Response 1 (*Egr1*), Activating Transcription Factor 3 (*ATF3*), Krüppel-Like Factor 4 (KLF4), Polo-Like Kinase 2 (*PLK2*) and Transforming Growth Factor Beta 3 (*TGFB3*); and some minimally expressed genes, were Sodium/Glucose Cotransporter 1 (*SGLT1/SLC5A1*), Peptide-transporters 1 (*PEPT1/SLC15A1*), Neutral And Basic Amino Acid Transporter (*NBAT/SLC3A1*), Monocarboxylate Transporter 3 (*MCT3/SLC16A8*), Iron-Regulated Transporter 1 (*IRT1/SLC40A1*), Mucin 13 (*MUC13*), Transferrin (*TF*), Wingless-Type MMTV Integration Site Family, Member 10B (*WNT10B*) and Bone Morphogenetic Protein 1 (*BMP1*), mainly involved in the regulation of cell division, gene transcription, signal transduction, gut development, nutrient transport and energy metabolism (Table 2). It is suggested that these genes might be related to IEC proliferation, differentiation. and maturation.

### 3.4. GO and KEGG Pathway Analyses of DEGs

To elucidate the correlation between DEGs and IEC renewal, GO and KEGG pathway analyses were then performed to analyze the enrichment functions and signaling pathways of the DEGs. Biological process GO enrichment analysis indicated that the DEGs were mainly involved in the cellular process (182 genes), biological regulation (172 genes), response to stimulus (145 genes), the multicellular organismal process (111 genes), the metabolic process (89 genes), the developmental process (86 genes), and signaling (78 genes). Cellular component GO enrichment analysis showed that the DEGs were mainly involved in the cell (203 genes), the membrane (168 genes), the membrane part (144 genes), the organelle (141 genes), the extracellular region (109 genes), and the organelle part (72 genes). Molecular function GO enrichment analysis showed that the DEGs were mainly involved in binding (226 genes), catalytic activity (106 genes), molecular function regulator (50 genes) and molecular transducer activity (39 genes) (Figure 3). In addition, KEGG pathway analysis revealed DEGs were categorized into 284 significant enriched pathways (Appendix A), and mainly enriched in signal transduction (Figure 4), of which 15 significant pathways involved in IECs renewal were showed in Figure 5. Importantly, Wnt, Hippo, TGF-beta, mTOR, PI3K-Akt, and MAPK signaling pathways were closely related to the differentiation, proliferation, maturation, and apoptosis of IECs. In particular, the number of DEGs enriched in the PI3K-Akt signaling pathway is the largest compared with other pathways.

### 3.5. Experimental Validation of DEGs Expression Using qRT-PCR

A total of eight DEGs (*HOXA5*, *ZFP36*, *KLF2*, *HSP70-2*, *MUC13*, *WNT10B*, *AQP10*, and *SLC15A1*) identified by RNA-seq were selected for validation via qRT-PCR. The qRT-PCR results indicated that the expression patterns of the eight genes between F3 and F1 corresponded well with the RNA-seq expression profiles (Figure 6). It also confirmed that the relative expression levels of the eight genes in F1 and F3 by qRT-PCR were ranged as significantly different, which was similar to those indicated by the RNA-seq analysis (Figure 7).

## 4. Discussion

The age of suckling is a critical stage of postnatal growth and gastrointestinal development in piglets. The early weaning of piglets (21-day weaning age) is accompanied by a series of environmental, psychological, and nutritional stresses, which cause poor growth, decreased feed intake and diarrhea [7]. Impairment in intestinal epithelial integrity and barrier function is a central etiological factor for ‘early weaning stress syndrome’ in piglets [10,11]. Thus, the balance of IEC proliferation, differentiation, migration, and apoptosis is critical for piglet gut epithelial health. In the present study, we performed RNA-seq analyses to identify piglet jejunal F1 and F3 genes that exhibited changes in expression at the suckling age of 21-days. Our results help to improve the understanding of the molecular mechanisms of intestinal epithelium renewal in suckling piglets on the basis of mRNA transcriptome profile analysis.

The IECs mainly represent the differentiated absorptive cell lineage (enterocytes) and play a key role in digesting and absorbing nutrient substance [2]. ALP is a brush border marker enzyme that is formed during maturation of enterocytes. Its activity may indicate the degree of cell differentiation to some extent. In the present study, we confirmed that the activity of ALP, lactase, and sucrase in F1 was significantly higher than in F3. Similarly, previous studies have demonstrated the activity of some digestive enzyme activities, key energy metabolic enzymes, and nutrient transporter gene expression in intestinal villus epithelial cells were significantly higher than in crypt cells in piglets [12,13]. It is well known that glucose is the main energy substance for gastrointestinal development, and its digestion and absorption in the IECs is the key to its utilization. As a glucose transporter, *SGLT1* is strongly expressed in enterocytes of the small intestine, and it is also highly expressed in crypt cells. The *PEPT1* knockout mouse model study showed that Gly-Sar dipeptide transport in IECs was inhibited, causing the influx of Ca^2+^ in enteroendocrine cells to be blocked, thereby reducing the secretion of glucagon-like peptide 1 (GLP-1) [14]. It confirmed that PEPT1 as a functional sensing and transport system for oligopeptides in the IECs in combination with the calcium-sensing receptor maintains intestinal homeostasis. Previous studies demonstrated that iron participates in the regulation of many important signal molecules in IECs and plays a key role in the formation of human intestinal villi [15]. Pu et al. [16] demonstrated that iron metabolism can promote the development of the intestine by improving its morphology in neonatal piglets, which maintains its mucosal integrity and enhances the expression of immuno-associated factors. Immunohistochemistry confirmed that the human MUC13 protein is not only expressed on the apical membrane of goblet cells in the gastrointestinal tract, but also within the goblet cell membrane [17]. It indicates that the expression of MUC13 is closely related to the formation of goblet cells. Our results revealed that nutrient transporters (e.g., *PEPT1*, *SGLT1*, and *TF*) and *MUC13* were strongly expressed in F1, in agreement with previous studies. Therefore, it suggested that the dramatical increase in those gene transcripts may be a crucial cause of induced ISC differentiation, and that it promoted the maturation of absorptive and secretory cell lineages in suckling piglets.

To understand the transcriptional mechanisms regulating the renewal of IECs along CVA, with reference to the relevant literature, we screened some key transcription factors and genes (*c-Jun*, *c-F**os*, *ATF3*, *KLF4*, *TGFB3*, *PLK2*, etc.) related to intestinal epithelial development. Activator Protein 1 (AP-1) is a dimeric transcription factor composed of *c-Jun*, *c-Fos*, or *ATF* (Activating Transcription Factor) subunits that bind to a common DNA site, the AP-1-binding site. The proto-oncogene c-Jun belongs to the AP-1 group of transcription factors, and it is a key regulator of ISC proliferation and tumorigenesis. Sancho et al. [18] reported that c-Jun was highly expressed in ISCs and the absence of c-Jun resulted in decreased IEC proliferation and villus height. Biteau et al. [19] found that c-FOS serves as a convergence point for EGF receptor signal and for the JNK signaling pathway, which promotes ISC proliferation and differentiation in response to stress. *ATF3* is strongly expressed in enterocytes, and tissue-specific knockdown *ATF3* in the fly gut leads to inhibit progenitor cell proliferation; furthermore, *ATF3* can also maintain normal intestinal barrier function via regulating JNK signal during acute infection [20]. Yu et al. [21] found a significant increase in goblet cell numbers, which affects the position of these cells in the small intestine CVA of *KLF4*−/− mice, suggesting that *KLF4* is not only crucial for goblet cell differentiation, but it is also required for maintaining the goblet cell population. *TGFB3* can mediate IEC apoptosis via regulating apoptosis-associated proteins Bcl-xL and Bcl-2 in *TGFB3* (+/−) heterozygous mice, which in turn causes a decrease in villus height [22]. McKaig et al. [23] revealed that the biologically active cytokine TGFB3 is mainly secreted by human colonic subepithelial myofibroblasts, which can enhance the recovery of wounded epithelial monolayers via a TGFB-dependent pathway. In the present study, transcription factors (*c-Jun*, *c-FOS*, *ATF3*, *KLF4*, and *TGFB3*) were significantly overexpressed in F3. This suggests that they may play a key role in maintaining stem cell properties and IEC differentiation. PLK2 phosphorylation is critical for Centrosomal P4.1-associated protein (CPAP) function in procentriole formation during the centrosome cycle, which is well known for maintaining centrosome and spindle integrity during cell division [24]. Here, the overexpression of *PLK2* in crypt cells is likely to indicate that it can regulate the cell cycle by promoting the proliferation of progenitor cells.

Furthermore, functional analysis reveals that some DEGs are enriched in related signaling pathways that regulate the renewal of IECs, including Wnt, TGF-beta, Hippo, FoxO, PI3K-Akt and MAPK signaling pathways. WNT10B are expressed in undifferentiated human embryonic stem cells and endoderm precursor cells [25]. WNT10B specifically activates the canonical Wnt/β-catenin signaling pathway and thus triggers β-catenin/TCF (T-cell-specific transcription factor)-mediated transcriptional programs in the nucleus [26]. Batlle et al. [27] confirmed that Wnt signals can also promote the maintenance of ISCs by regulating the EphB-Ephrin B signaling gradient along the CVA to form different cell lineages in the intestinal epithelium. Wnt signaling is crucial for the lineage specification of ISCs, especially the differentiation of Paneth cells; moreover, activation of Wnt signaling drives the formation of massive ectopic Paneth cells. Recent studies suggest that Wnt signaling can also regulate the differentiation of enteroendocrine cells and goblet cells. BMP1 plays an essential role in cell differentiation and morphogenesis in animal embryos via activating related signaling pathways and growth factors. Ryuichi et al. [28] reported that BMP1 can physically bind to other BMPs, and thus activate other BMPs by this interaction. BMPs belong to the TGF-beta superfamily, which can inhibit ISC activation and expansion, promoting the terminal differentiation of several secretory lineages, including enteroendocrine cells, Paneth cells and goblet cells [29,30]. Snippert et al. [31] reported that intestinal crypt homeostasis results from neutral competition between symmetrically dividing Lgr5+-expressing ISCs, which are essential during IEC regeneration. HOXA5, as an important repressor of ISCs fate, forms negative feedback with the Wnt signaling to maintain stemness, and becomes active in the intestinal villi area where it inhibits Wnt signaling to generate all intestinal epithelial lineages [32]. Imajo et al. [33] found that Hippo signalling has a dual role in the renewal of CVA through the regulation of ISC proliferation and differentiation into a specific secretory lineage via two effectors of Hippo signalling, Yes-associated protein 1 (YAP) and transcriptional co-activator with PDZ-binding motif (TAZ). Ding et al. [34] revealed that alterations in MAPKs (i.e., inhibition of ERK activity and induction of JNK expression) can promote differentiation and apoptosis in the human colon cancer cell lines Caco-2 and HT29. Activating JNK signalling can increase the number of IECs and the villus height by regulating the proliferation and migration of progenitor cells [18]. Zhang et al. [35] reported that mitochondria modulate proliferation of ISCs and induce secretory cell lineage differentiation through the Forkhead box O (FOXO) signaling pathway. Here, our results were consistent with those of previous studies, and it is worth noting that most DEGs were enriched in the PI3K-Akt signaling pathway, which is known to control critical cellular processes by regulating apoptosis, protein synthesis, metabolism, and the cell cycle [36]. It indicates that the renewal of the intestinal epithelial CVA cells is a complex molecular signal regulation network. More work is needed in the future to elucidate the exact mechanisms of ISC differentiation, IEC proliferation and apoptosis in suckling piglets.

In addition, Xiong et al. [7] reported that porcine IECs have notably high energy demands due to the rapid renewal of the intestinal epithelium within a few days. In the present study, our results have shown that some DEGs enriched to mTOR and AMPK signal pathway. mTOR exists in two complexes (mTORC1 and mTORC2) and regulates cell growth, proliferation and metabolic processes [37]. As a sensor of nutrition signals, mTORC1 can sense the signals of amino acids, growth factors, glucose, and other signals in the microenvironment, and then regulate IEC energy metabolism including glycolysis and oxidative phosphorylation [38]. Setiawan et al. [39] demonstrated that the constitutive activation of mTORC1 can increase proliferation and faster migration of IECs to villi, controlling at the same time the correct clustering and migration of Paneth cells in the small intestine. AMP-activated protein kinase, as a master regulator of energy metabolism, can impair intestinal barrier function and IEC differentiation via regulating caudal type homeobox 2 (CDX2) expression [40]. Hence, our data indicate that energy metabolism is essential in maintaining the piglet IEC renewal process. Based on this, further studies should be conducted to improve the energy supply to piglets, thus improving intestinal function.

## 5. Conclusions

In the present study, we successfully isolated the intestinal upper villus epithelial cells (F1) and crypt epithelial cells (F3) from the jejunum of 21-day suckling piglets by the chelation precipitation technique. Our data indicated that the ALP and disaccharidase activities of F1 were significantly higher than those of F3. Cluster analysis of the DEGs between F1 and F3 identified a list of genes, of which 448 were more highly expressed in F1 and 224 were overexpressed in F3. Most DGEs were reported to be associated with IEC proliferation, differentiation, apoptosis, and intestinal function. The DGEs were significantly associated with 284 KEGG pathways, such as mTOR, TGF-beta, Wnt, and MAPK pathways, which are known to play key roles in signaling communication in intestinal epithelium renewal. In addition, our results found that most DEGs were enriched in the PI3K-Akt signaling pathway. All in all, the molecular mechanism of cell renewal along the intestinal epithelium CVA of suckling piglets reported in this study provides a potential new direction for gene regulation and signal transduction to repair IECs of piglets via nutritional intervention.

## Figures and Tables

**Figure 1 animals-12-02324-f001:**
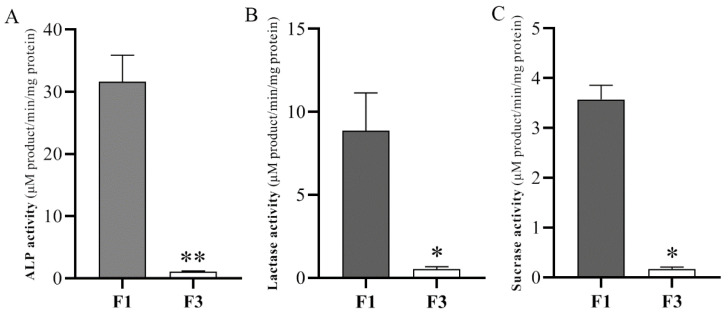
Relative enrichments of enzymes in the intestinal upper villus epithelial cells (F1) and crypt epithelial cells (F3) isolated along the small intestinal crypt-villus axis of 21-day sucking piglets. Enzymes: alkaline phosphatase (**A**), lactase (**B**), and sucrose (**C**) specific activities. Data with error bars represents means ± SEM, n = 5 for each group. * *p* < 0.05, ** *p* < 0.01.

**Figure 2 animals-12-02324-f002:**
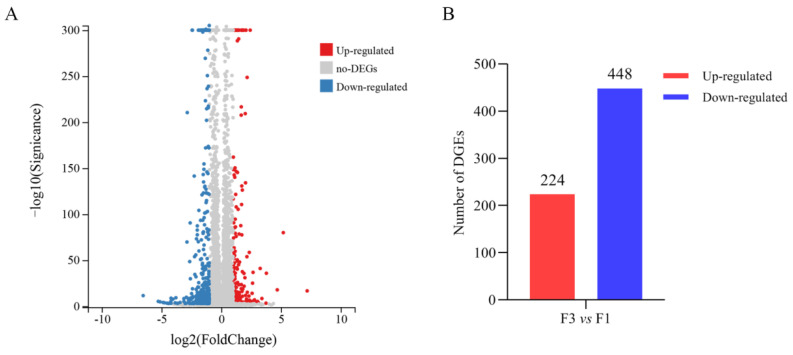
(**A**) Volcano map of the library size normalized sequencing read counts data from the two groups showing statistically DEGs. A dot represents a gene, while color represents whether they are significantly highly expressed (red) or minimally expressed (blue). (**B**) Statistics of the number of DEGs. The *x*-axis represents the control-case difference pair, and the *y*-axis represents the number of differential genes. Red and blue represent the total number of highly expressed and minimally expressed DEGs, respectively. vs is abbreviation of versus.

**Figure 3 animals-12-02324-f003:**
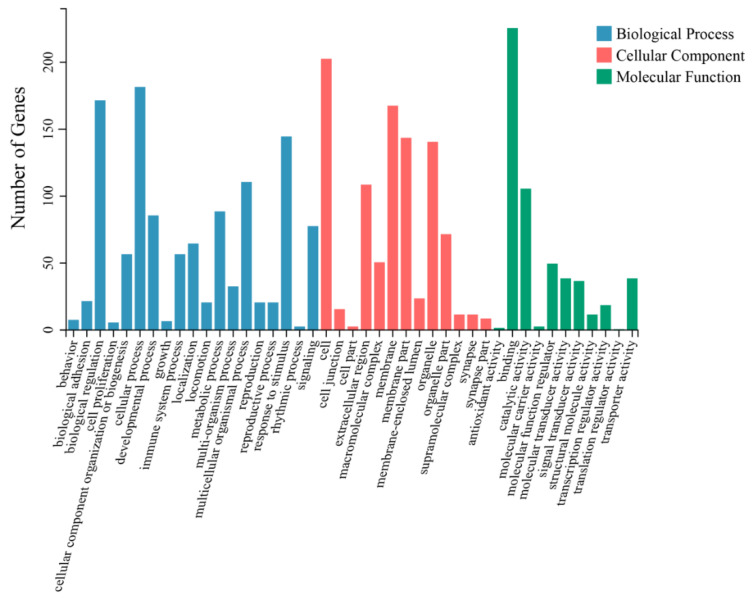
Gene Ontology (GO) functional analysis of unique sequences (DEGs) from the crypt epithelial cells (F3) vs. the villus upper epithelial cells (F1) transcriptome. Unique sequences (DEGs) were assigned to three categories: molecular functions, biological processes and cellular components.

**Figure 4 animals-12-02324-f004:**
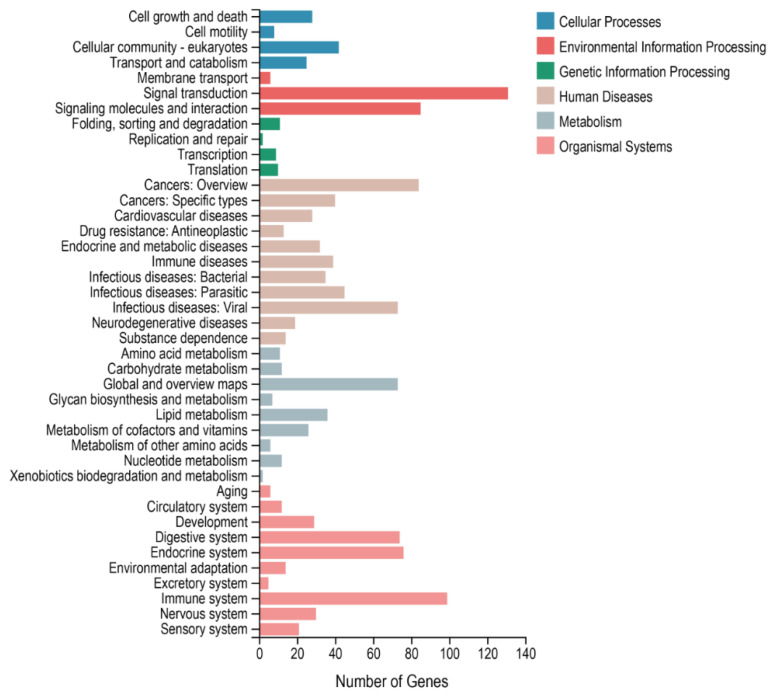
Significantly enriched KEGG pathways of DEGs from the crypt epithelial cells (F3) vs. the villus upper epithelial cells (F1) transcriptome. The phyper function in R software version 1.10.0 (AT&T Bell Laboratories, New York, NY, USA) was used for enrichment analysis, and then FDR correction was performed for the *p* value to obtain a Q value. The function of the Q value ≤ 0.05 is regarded as significant enrichment.

**Figure 5 animals-12-02324-f005:**
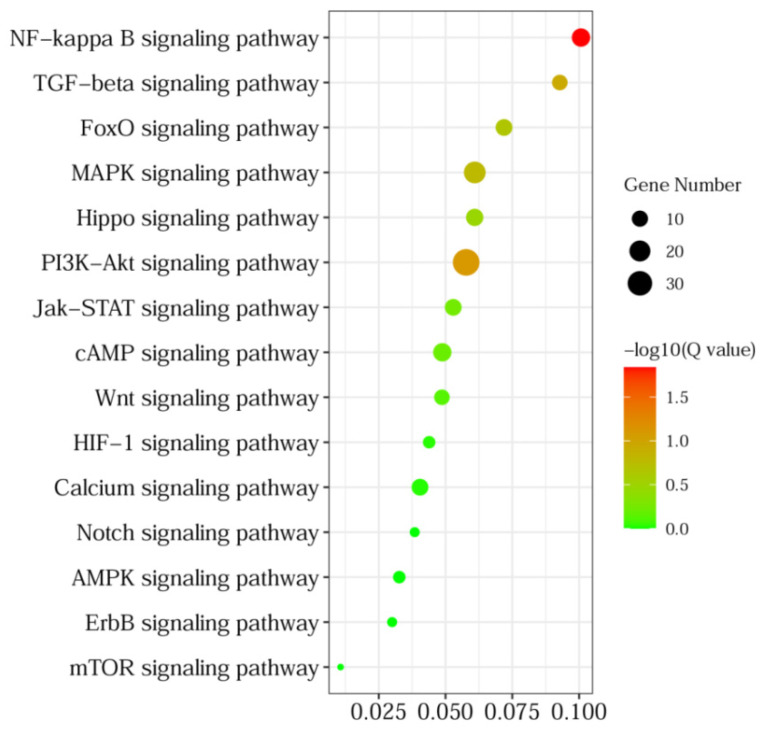
Significantly enriched KEGG pathways of the DEGs (the crypt epithelial cells (F3) vs. the villus upper epithelial cells (F1)) belonging to signal transduction.

**Figure 6 animals-12-02324-f006:**
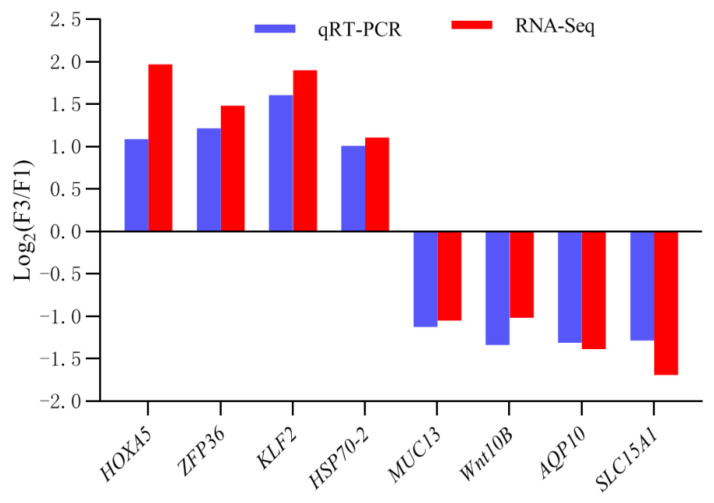
Comparison of the DEGs expression level of RNA-seq with qRT-PCR results between the crypt epithelial cells (F3) and the villus upper epithelial cells (F1).

**Figure 7 animals-12-02324-f007:**
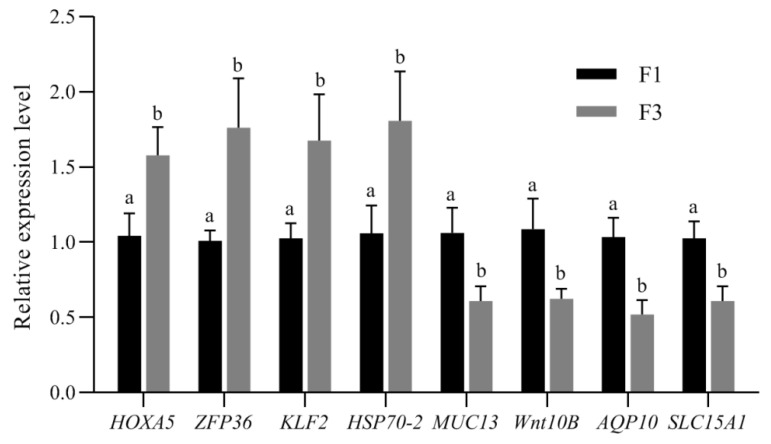
Analysis of the eight DEGs relative expression level by qRT-PCR. *HOXA5*, Homeobox A5; *ZFP36*, Zinc Finger Protein 36 (Tristetraprolin, TTP); *KLF2*, Krüppel-Like Factor 2; *HSP70-2*, Heat Shock Protein Family A (Hsp70) Member 2; *MUC13*, Mucin 13; *WNT10B*, Wingless-Type MMTV Integration Site Family, Member 10B; *AQP10*, Aquaporin 10; and *SLC15A1*, Peptide-transporters 1. In each panel, different lowercase letters (a and b) indicate significant differences between the crypt epithelial cells (F3) and the villus upper epithelial cells (F1). (Value are means ± SEM and expressed as arbitrary units (normally expression is 1 arbitrary unit relative to GAPDH gene expression). *p* < 0.05, n = 5 for each group).

**Table 1 animals-12-02324-t001:** Statistics for filtering and mapping reads in each library.

Sample Name	F1 ^a^	F3 ^b^
Raw Reads	23,957,221	23,957,192
Q20 ^c^	93.90	96.00
Clean Reads	23,881,637	23,894,452
Total Mapped Reads (%)	88.35	90.21
Unique Match (%)	57.16	63.18
Multi-position Match (%)	31.19	27.03
Total Unmapped Reads (%)	11.65	9.80

^a^ F1: the villus upper epithelial cells; ^b^ F3: the crypt epithelial cells; ^c^ Q20: the percentage of bases whose mass value is greater than or equal to 20.

**Table 2 animals-12-02324-t002:** Summary of the 18 DEGs related to crypt-villus axis renewal.

Gene ID	Symbol	Description (Gene Name)	Log2 Ratio(F3/F1) ^a^^b^	FDR ^c^
397113	*SLC5A1*	Sodium/Glucose Cotransporter 1 (*SGLT1*)	−1.149	0
397624	*SLC15A1*	Peptide-transporters 1 (*PEPT1*)	−1.691	0
100144588	*SLC3A1*	Neutral And Basic Amino Acid Transporter (*NBAT*)	−1.385	5.17 × 10^−3^
100520648	*SLC16A8*	Monocarboxylate Transporter 3 (*MCT3*)	−1.765	8.19 × 10^−4^
100737517	*SLC40A1*	Iron-Regulated Transporter 1 (*IRT1*)	−1.144	0
396845	*TF*	Transferrin	−1.222	1.465 × 10^−20^
100125829	*MUC13*	Mucin 13	−1.050	0
100126276	*WNT10B*	Wingless-Type MMTV Integration Site Family, Member 10B	−1.015	3.028 × 10^−6^
100156461	*BMP1*	Bone Morphogenetic Protein 1	−1.915	0
100151994	*LGR5*	Leucine Rich Repeat Containing G Protein-Coupled Receptor 5	1.031	8.174 × 10^−30^
397423	*HOXA5*	Homeobox A5	1.969	2.04 × 10^−5^
396913	*JUN*	Jun Proto-Oncogene, AP-1 Transcription Factor Subunit (*c-Jun*)	1.362	0
100144486	*FOS*	Fos Proto-Oncogene, AP-1 Transcription Factor Subunit (*c-Fos*)	1.769	0
100520726	*EGR1*	Early Growth Response 1	2.074	0
100738612	*ATF3*	Activating Transcription Factor 3	1.875	0
595111	*KLF4*	Krüppel-Like Factor 4	1.161	0
397251	*PLK2*	Polo-Like Kinase 2	2.058	0
397400	*TGFB3*	Transforming Growth Factor Beta 3	1.607	7.343 × 10^−7^

^a^ F1: Reads count in the villus upper epithelial cells; F3: Reads count in the crypt epithelial cells. ^b^ Negative/(−)Log2 Ratio represent decreased levels of gene expression. Positive/(+) Log2 Ratio represent increased expression levels of gene expression. ^c^ FDR: False discovery rate, which was used to determine the *p*-value threshold in multiple tests.

## Data Availability

The datasets analyzed in the current study are available from the corresponding author on reasonable request.

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
