# Peer review of "Transcriptome Profile Analysis of Intestinal Upper Villus Epithelial Cells and Crypt Epithelial Cells of Suckling Piglets"

_animals, 2022, doi:10.3390/ani12182324_

Round 1
Reviewer 1 Report
Comments to the Authors:
I have read and reviewed your manuscript entitled " Transcriptome Profile Analysis of Intestinal Upper Villus Epithelial Cells and Crypt Epithelial Cells of Suckling Piglets ". In my opinion, the Authors present valuable data, however, the present form of the paper does not fit for publication in the Animals journal. The list of my comments and suggestions is presented below:
1. Chapter 2.2 “Measurement of alkaline phosphatase, Lactase, Sucrase, and Maltase Activities”: in this chapter there is no information about assay kit for maltase activity measuring and in chapter 3.1. there are no results of maltase activity.
2. Lines 106-117: did Authors use porcine assay kit? What method of protein measurement was used?
3. Chapter 2.7. This chapter does not follow the MIQE guidelines for publication of qPCR results (primers concentration, method thermal profile and so on). Have you performed primers validation?
4. Lines 178-179 there should be “sucrase” instead of “sucrose”
5. Fig. 7 there is no unit (I suggest the use of “arbitrary units”)
6. Whole manuscript is a bit chaotic and messy. There is a lot of editorial errors, i.e. “p”/”P”, line 240: incomplete sentence “… up-regulated (red) or down-regulated.”, gene names with italic/non-italic, etc.
Author Response
Response to Reviewer 1 Comments
Comments and Suggestions for Authors
I have read and reviewed your manuscript entitled "Transcriptome Profile Analysis of Intestinal Upper Villus Epithelial Cells and Crypt Epithelial Cells of Suckling Piglets". In my opinion, the Authors present valuable data, however, the present form of the paper does not fit for publication in the Animals journal. The list of my comments and suggestions is presented below:
1. Chapter 2.2 “Measurement of alkaline phosphatase, Lactase, Sucrase, and Maltase Activities”: in this chapter there is no information about assay kit for maltase activity measuring and in chapter 3.1.there are no results of maltase activity.
Response 1: Thank you for the good comments. Correction has been made in the revised version (Chapter 2.2 and 3.1).
2. Lines 106-117: did Authors use porcine assay kit? What method of protein measurement was used?
Response 2: Thank you for the good comments. Correction has been made in the revised version (Chapter 2.2).
3. Chapter 2.7. This chapter does not follow the MIQE guidelines for publication of qPCR results (primers concentration, method thermal profile and so on). Have you performed primers validation?
Response 3: Thank you for the good comments. Correction has been made in the revised version (Chapter 2.7).
4. Lines 178-179 there should be “sucrase” instead of “sucrose”
Response 4: Thank you for the good comments. Correction has been made in the revised version (Fig. 1 legend).
5.Fig. 7 there is no unit (I suggest the use of “arbitrary units”)
Response 5: Thank you for the good comments. Correction has been made in the revised version (Fig. 7 legend).
6.Whole manuscript is a bit chaotic and messy. There is a lot of editorial errors, i.e. “p”/”P”, line 240: incomplete sentence “… up-regulated (red) or down-regulated.”, gene names with italic/non-italic, etc.
Response 6: Thank you for the good comments. Correction has been made in the revised version.

Reviewer 2 Report
This study utilized RNA-seq technique to analyze and distinguish the different mRNA expression levels in crypt and villus cells of suckling piglet jejunum. Although this study seems interesting, the manuscript is very difficult to understand. Since this study is identifying quantitatively different expression level of various transcripts in crypt and villus cells, it should be indicated as highly expressed and minimally expressed transcript level in different cell types, rather than using up-regulated and down-regulated transcript level. In addition, this manuscript also needs English editing.
Specific comments:
1. line-11: change “investigated” to investigate
2. line-27: “change chelating precipitation technology” to divalent chelation and precipitation technique
3. line-31: where the unigenes are up- and down-regulated?
4. line-32: define what is DEGs
5. line-52: mammalian to mammals
6. line-98: provide the composition of chelating buffer.
7. Line-111: don’t understand what “supernatants were analysis”
8. Line-113: indicate the specific parameters of enzyme activities expressed in the results
9. Line-115: What is preparation?
10. Line-112: how the cDNA library was contracted?
11. Line-138: the sentence is ugly
12. Line-151: what Is gDNA?
13. Line-170: provide reference
14. Figure 1: provide unit of each enzyme activity in y-axis
15. Line-178: what is mark enzyme?
16. Line-218: “224 were upregulated” – it is not upregulated, it is highly expressed
17. Line-219: “448 were down-regulated” – it is not down-regulated, it is minimally expressed or expressed in low level
18. Lines-240 to 242: up-regulated /down-regulated – see comment #16&17
19. Lines 225-256: the sentence does not make any sense
20. Lines-295 to 299: What is (A), (B) etc… (H) referred to?
21. Figure 7: what that “a” and “b” next to the error bars denote?
22. Lin-305: what is age of sucking?
Author Response
Response to Reviewer 2 Comments
Comments and Suggestions for Authors
This study utilized RNA-seq technique to analyze and distinguish the different mRNA expression levels in crypt and villus cells of suckling piglet jejunum. Although this study seems interesting, the manuscript is very difficult to understand. Since this study is identifying quantitatively different expression level of various transcripts in crypt and villus cells, it should be indicated as highly expressed and minimally expressed transcript level in different cell types, rather than using up-regulated and down-regulated transcript level. In addition, this manuscript also needs English editing.
Specific comments:
- line-11: change “investigated” to investigate
Response 1: Thank you for the good comments. Correction has been made in the revised version (Chapter Simple Summary).
- line-27: “change chelating precipitation technology” to divalent chelation and precipitation technique
Response 2: Thank you for the good comments. Correction has been made in the revised version (Chapter Abstract).
- line-31: where the unigenes are up- and down-regulated?
Response 3: Thank you for the good comments. Correction has been made in the revised version (Chapter Abstract).
- line-32: define what is DEGs
Response 4: Thank you for the good comments. Correction has been made in the revised version (Chapter Abstract).
- line-52: mammalian to mammals
Response 5: Thank you for the good comments. Correction has been made in the revised version (Chapter 1. Introduction).
- line-98: provide the composition of chelating buffer.
Response 6: Thank you for the good comments. Correction has been made in the revised version (Chapter 2.1).
- Line-111: don’t understand what “supernatants were analysis”
Response 7: Thank you for the good comments. Correction has been made in the revised version (Chapter 2.2).
- Line-113: indicate the specific parameters of enzyme activities expressed in the results
Response 8: Thank you for the good comments. Correction has been made in the revised version (Chapter 2.2).
- Line-115: What is preparation?
Response 9: Thank you for the good comments. Correction has been made in the revised version (Chapter 2.3).
- Line-112: how the cDNA library was contracted?
Response 10: Thank you for the good comments. Correction has been made in the revised version (Chapter 2.4).
- Line-138: the sentence is ugly
Response 11: Thank you for the good comments. Correction has been made in the revised version (Chapter 2.5).
- Line-151: what Is gDNA?
Response 12: Thank you for the good comments. Correction has been made in the revised version (Chapter 2.7).
- Line-170: provide reference
Response 13: Thank you for the good comments. Correction has been made in the revised version (Chapter 3.1).
- Figure 1: provide unit of each enzyme activity in y-axis
Response 14: Thank you for the good comments. Enzyme activity is unit/g(mg) protein, which has been shown on the Y-axis.
- Line-178: what is mark enzyme?
Response 15: Thank you for the good comments. Correction has been made in the revised version (Fig. 1 legend).
- Line-218: “224 were upregulated” – it is not upregulated, it is highly expressed
Response 16: Thank you for the good comments. Correction has been made in the revised version (Chapter 3.3).
- Line-219: “448 were down-regulated” – it is not down-regulated, it is minimally expressed or expressed in low level
Response 17: Thank you for the good comments. Correction has been made in the revised version (Chapter 3.3).
- Lines-240 to 242: up-regulated /down-regulated – see comment #16&17
Response 18: Thank you for the good comments. Correction has been made in the revised version (Fig. 2 legend).
- Lines 225-256: the sentence does not make any sense
Response 19: Thank you for the good comments. Correction has been made in the revised version (Chapter 3.4).
- Lines-295 to 299: What is (A), (B) etc… (H) referred to?
Response 20: Thank you for the good comments. Correction has been made in the revised version (Fig. 7 legend).
- Figure 7: what that “a” and “b” next to the error bars denote?
Response 21: Thank you for the good comments. Correction has been made in the revised version (Fig. 7 legend).
- Line-305: what is age of sucking?
Response 22: Thank you for the good comments. Correction has been made in the revised version (Chapter 4. Discussion).

Reviewer 3 Report
Suggestions, comments and corrections were included in the attached pdf file.
Please verify carefully the English writing.
Please improve/correct some figure presentation and legend in order to have better results presentation
Verify some sentences al over the text that are not clear and distract from fluent reading

Author Response
Response to Reviewer 3 Comments
Comments and Suggestions for Authors
Suggestions, comments and corrections were included in the attached pdf file.
Please verify carefully the English writing.
Please improve/correct some figure presentation and legend in order to have better results presentation
Verify some sentences al over the text that are not clear and distract from fluent reading.
Specific comments:
- line-11: change “investigated” to investigate
Response 1: Thank you for the good comments. Correction has been made in the revised version (Chapter Simple Summary).
- line-11: change “drives” to drive
Response 2: Thank you for the good comments. Correction has been made in the revised version (Chapter Simple Summary).
- line-19: Is this verb necessary in this sentence? Is not clear the sentence as it is
Response 3: Thank you for the good comments. Correction has been made in the revised version (Chapter Simple Summary).
- line-36: What GO stand for? Have to write the complete name as it is the first time that appears in the text (is that gene ontology?)
Response 4: Thank you for the good comments. Correction has been made in the revised version (Chapter Abstract).
- line-58: Paneth cells are located at the crypt level and not in the villus
Response 5: Thank you for the good comments. Correction has been made in the revised version (Chapter 1. Introduction).
- lines-73 to 74: delete “weaning” and “chosen”
Response 6: Thank you for the good comments. Correction has been made in the revised version (Chapter 1. Introduction).
- Line-101: Which suspension is that (is that a commercial one? Put the company where adquired or the composition if it was made by the authors
Response 7: Thank you for the good comments. Correction has been made in the revised version (Chapter 2.1).
- Line-131: change “is” to was
Response 8: Thank you for the good comments. Correction has been made in the revised version (Chapter 2.5).
- Line-134: not clear; it is redundant
Response 9: Thank you for the good comments. Correction has been made in the revised version (Chapter 2.5).
- Line-145: 1.10.0 is year?
Response 10: Thank you for the good comments. Correction has been made in the revised version (Chapter 2.6).
- Line-159: Why not the Standard Deviation-SD
Response 11: Thank you for the good comments. The objective of the present study was mainly to compare the mean sizes of the two groups of samples, so SEM was chosen.
- Line-162: To compare what vs. what? Students’s t-test is used to compare 2 means. In this case wihich group were you comparing?
Response 12: Thank you for the good comments. In the present study, we compare enzyme activity and the relative expression levels of DEGs in F1 group and F3 group by Students’s t-test. Correction has been made in the revised version (Chapter 2.8).
- Line-167: delete “In the present study”
Response 13: Thank you for the good comments. Correction has been made in the revised version (Chapter 3.1).
- Line-170: reference?
Response 14: Thank you for the good comments. Correction has been made in the revised version (Chapter 3.1).
- Lines-172 to 173: Re-write, it is not clear as it is
Response 15: Thank you for the good comments. Correction has been made in the revised version (Chapter 3.1).
- Line-188: not clear, looks like incomplete sentence
Response 16: Thank you for the good comments. Correction has been made in the revised version (Chapter 3.2).
- Lines 189-190: for F1 and F3, respectively? Those value are not in Table 1, the text is not clear. Re-write please.
Response 17: Thank you for the good comments. Correction has been made in the revised version (Chapter 3.2).
- Line-214: delete “of suckling piglets”
Response 18: Thank you for the good comments. Correction has been made in the revised version (Chapter 3.3).
- Lines-220 to 225: change “such as” to “were”
Response 19: Thank you for the good comments. Correction has been made in the revised version (Chapter 3.3).
- Line-231: delete “were”
Response 20: Thank you for the good comments. Correction has been made in the revised version (Chapter 3.3).
- Line-233: change “suggesting” to “suggested”
Response 21: Thank you for the good comments. Correction has been made in the revised version (Fig. 7 legend).
- Fig 2B: Is this x-axis name corrected? If yes, which is F1 and which is F3? Why is not explained in the Fig 2 legend, maybe the correct name is “Gene expression”.
Response 22: Thank you for the good comments. Correction has been made in the revised version (Fig 2 legend).
- Table 2: It is suggested to include a table 2 legend to better explain the (-) negative Log2 Ratio
Response 23: Thank you for the good comments. Correction has been made in the revised version (Table 2 legend).
- Lines-252 to 256: not clear, improve writing
Response 24: Thank you for the good comments. Correction has been made in the revised version (Chapter 3.4).
- Line-261: delete “in signal transduction in the present study”
Response 25: Thank you for the good comments. Correction has been made in the revised version (Chapter 3.4).
- Line-269: Where in the Fig 3 depicted which DEG are from F1 and which from F3.
Response 26: Thank you for the good comments. Correction has been made in the revised version (Fig 3 legend).
- Line-274: in which group? F1 or F3
Response 27: Thank you for the good comments. Correction has been made in the revised version (Fig 4 legend).
- Line-276: This is not evident in the Fig 4. Where is the Q value? It is seen in Fig 5.
Response 28: Thank you for the good comments. The significant enrichment of DEGs into the KEGG pathway is based on the Q value≤0.05. Correction has been made in the revised version (Fig 4 legend).
- Line-279: Where? In F1 or F3? change “belonged” to “belonging”
Response 29: Thank you for the good comments. Correction has been made in the revised version (Fig 5 legend).
- Line-287: change “from” to “as”
Response 30: Thank you for the good comments. Correction has been made in the revised version (Chapter 3.5).
- Line-305: change “sucking” to “suckling”
Response 31: Thank you for the good comments. Correction has been made in the revised version (Chapter 4. Discussion).
- Line-306: change “a series” to “by a series”
Response 32: Thank you for the good comments. Correction has been made in the revised version (Chapter 4. Discussion).
- Line-309: delete “which”
Response 33: Thank you for the good comments. Correction has been made in the revised version (Chapter 4. Discussion).
- Line-324: change “knows” to “known”
Response 34: Thank you for the good comments. Correction has been made in the revised version (Chapter 4. Discussion).
- Line-362: change “also” to “it is also”
Response 35: Thank you for the good comments. Correction has been made in the revised version (Chapter 4. Discussion).
- Line-374: change “be involved in” to “by”
Response 36: Thank you for the good comments. Correction has been made in the revised version (Chapter 4. Discussion).
- Line-397: delete “which”
Response 37: Thank you for the good comments. Correction has been made in the revised version (Chapter 4. Discussion).
- Lines 405 to 408: not a clear sentence
Response 38: Thank you for the good comments. Correction has been made in the revised version (Chapter 4. Discussion).
39: Line-419: delete “It will knows that”
Response 39: Thank you for the good comments. Correction has been made in the revised version (Chapter 4. Discussion).
- Line-425: change “controlled” to “controlling”
Response 40: Thank you for the good comments. Correction has been made in the revised version (Chapter 4. Discussion).
- Line-440: change “some of which the” to “such as”
Response 41: Thank you for the good comments. Correction has been made in the revised version (Chapter 5. Conclusions).
- Line-441: delete “are”
Response 42: Thank you for the good comments. Correction has been made in the revised version (Chapter 5. Conclusions).
- Lines 447 to 448: not clear
Response 43: Thank you for the good comments. Correction has been made in the revised version (Chapter 5. Conclusions).

Round 2
Reviewer 2 Report
Line-137: Please indicate what is measured for every enzyme activity (e.g., ug or mg Pi released/g protein for alkaline phosphatase). Also provide the unit of activity measured instead of giving U/g protein in the Y-axis of Figure 1.
Line-256: Change "Eenzymes" to "Enzymes"
Please check the Spelling through out.
Author Response
Response to Reviewer 2 Comments
Comments and Suggestions for Authors
- Line-137: Please indicate what is measured for every enzyme activity (e.g., ug or mg Pi released/g protein for alkaline phosphatase). Also provide the unit of activity measured instead of giving U/g protein in the Y-axis of Figure 1.
Response 1: Thank you for the good comments. Correction has been made in the revised version (Chapter 2.2 and Fig 1).
- Line-256: Change "Eenzymes" to "Enzymes"
Response 2: Thank you for the good comments. Correction has been made in the revised version (Fig 1 legend).
- Please check the Spelling through out.
Response 3: Thank you for the good comments. Correction has been made in the revised version.
